# Digital Privacy Transparency, a Meta Privacy Model for Glass-boxed Data Governance

Mark Lizar,

[1] *Global Privacy Rights, Montreal, Quebec, Canada*

### Abstract

This paper introduces this Digital Privacy Transparency Standard, meta-privacy model to address the international semantic and data governance interoperability challenges faced currently with data protection focused notice record and consent receipt information structures.

This meta-model for digital privacy transparency introduces Glassbox data governance structure that operates much like banking transactions with accounts that track and logs transaction events with records and receipts. But, rather that Glassbox governance for currency, standard transparency provides Glassbox governance for data, wherin standard notice record and receipts information structure is used to govern across 3 vectors of data control.

1.  Personal Data Control – Consent based data governance model
2.  Data Protection – the current regulated data governance model
3.  Co-Regulated data control, representing a system of publicly recorded notices using international open and free to access ISO/IEC standards, suitable for public digital transparency infrastructure.

This introduces a standard digital privacy notice record and consent receipt assurance and exchange information structure that can be applied to all 3 vectors of data control governance, in all legal jurisdictions, for all stakeholders, used for all physical (iot) and digital surveillance devices, as well as meeting the hight Canadian standard for consent.

Introducing 0PN Glassbox Data Governance model for internationally adequate notice and digital consent.

### Keywords

Digital Privacy, Privacy Meta-model, Standard Digital Privacy Pattern,  Glassbox data governance,

## 1.  Introduction

Transparency, notice and record of notice is required for all human technical interaction which is why this is references as a meta-privacy model, and as a result ideal to be standardised internationally, to support an international notice record and receipt system,  in which a record is generated for all legitimate  surveillance and data processing activates.

Taking into account,  WP29 Guidance found within the first  5 recitals of this guidance, it clear that operationally, transparency is a foundational privacy principal, relevant to all data protection regulating defined data processing.

WP 29 Transparency Guidelines 2017

| Recital 1 | **Transparency is an overarching obligation** under the GDPR applying to three central areas: |
| --- | --- |
| | .. as compliance with transparency is required in relation to data processing under Directive (EU) 2016/6803, these guidelines also apply to the interpretation of that (transparency) principle. |
| Recital 2 |  in addition to the requirements that data must be processed lawfully and fairly, **transparency is now included as a fundamental aspect of these principles.** |
| Recital 4. | The concept of transparency in the GDPR is user-centric rather than legalistic |

The OPN Digital Privacy Transparency model prioritises the principle of transparency, openness, choice  to implement  international transparency standard, referring to notice record and receipt standard in accordance to CoE convention 108+, utilising the EU GPR, (international vs of the GDPR) as guidance for lawful access, or what is referred to as operational (access) to personal data held by Data Controllers. (Article 70 – 94 found in Appendix section 7.2)

This year, the Council of Europe's Convention 108+ finally ratified, (March 2025) and expected to soon come fully into force once 38 countries have passed legislation to enforce it.

These customs, supported by common law have developed over time  through  consensus, and common rule development dating back to the Magna Carta and Forrest Charter 1217. Evolving from a Charter, into common laws, extended by principles and their best practices, into constitutions.

## 2. Three Vectors of Data Governance

The operational privacy notice, privacy pattern, schema and consent record information structure presented here introduces a 3 vectors of data governance transparency for integrating human data control, exchange and governance interoperability into existing data flows, policies and frameworks.  Defined in accordance to the international Convention 108+ Treaty, and the EU international version of the GDPR, to enable dynamic, real time lawful access, to address growing cybersecurity vulnerabilities related to a 'data protection' only semantic governance model.

### 2.1.1.    Data Protection Legislation and Policy

Data protection, like the EU GDPR, and CCPA regulation are updates to legislation that was first written in the 1970's before the internet when consent was always  physical interaction with analogue and written consent where notice and identification was inherent to the context.  Often requiring a human signature in front of the Controller of this personal data and the meta data about this interaction.  Policy driven forward by the 1967 Helsinki declaration[1] for explicit consent in health care.

As a result transparency, referred to generally as notice, and the legal basis of consent is specified in legislation for peer to peer brick and mortar analogue privacy contexts.  Wherein an individual must identify themselves, prior to the Controller to access privacy rights.

In analogue privacy context Controllers are almost always the Data Controller, where both parties identify themselves at the same time often under the premise or assumption of operating in common

---

[1] World Medical Association. Declaration of Helsinki: Ethical Principles for Medical Research Involving Human Subjects. Adopted by the 18th WMA General Assembly, Helsinki, Finland, June 1964. -

public space where the expected data privacy access permission of the context and purpose of use will be respected.

In a physical, peer to peer context A  written and explicit consent is given in person without any intermediaries, often including the required permissions in what is referred to as a consent form which can be assured directly by the PII Controller.   Often containing 'opt-in' check boxes that can be  ticked, to set these permissions inf the context of surveillance the consent is for.

This is fundamentally different then online, where an invisible 3rd party Controller or processor may be involved, from another country or territory. A challenge not directly addressed in data protection regulation.  Making it difficult if not impossible to track subsequent access, processing and use of PII once shared.
.

## 2.2.    OPN-Governance Policy Instrument Stack

This model utilises the transparency requirements and modalities defined by the appropriate international instruments to be a applicable for all digital privacy, identification and surveillance contexts, so as to scale as a solution for all systems that require common data governance transparency rules,

### 2.2.1.        Normative

- 0PN- Glassbox Model, Transparency Policy and Notice Semantics,
  - transparency by design with the clear and plain language being written and presented from the context of consent, by default, (rather than presented according to data protection semantics, which is notice written using  Data Protection Language meant to govern analogue privacy and consent contexts.
- Convention 108+ :
  - as the standard privacy policy, it provides the authoritative code of conduct for international adequacy between countries, and legal jurisdictions, applicable to all stakeholders, in all safety, security and privacy contexts,
  - Is the authoritative international Code of Transparency Conduct
- EU DPR (Data Protection Regulation)[2]  for lawful access
  - Provides the co-regulatory privacy policy best practices for lawful access, or what is referred to as operational personal data held by PII controllers
  - Authoritative international code of practice for lawful access with consent or a legitimate basis in law.
- ISO/IEC 29100 security and privacy framework
- ISO/IEC 27560 consent record information structure with the Candian-International Notice record and receipt record information structure

### 2.2.2.        Non-Normative

- W3C Data Privacy Vocabulary

---

[2] EU DPR (Data Protection Regulation)  is largely  the mirror of the GDPR but it also includes Chapter IX, Art 70-91 for PROCESSING OF OPERATIONAL PERSONAL DATA BY UNION BODIES, OFFICES AND AGENCIES
Operational Personal Data, in the REGULATION (EU) 2018/1725 OF THE EUROPEAN PARLIAMENT AND OF THE COUNCIL of 23 October 2018, on the protection of natural persons with regard to the processing of personal data by the Union institutions, bodies, offices and agencies and on the free movement of such data, and repealing Regulation (EC) No 45/2001 and Decision No 1247/2002/EC

- o Vector 1, policy semantics are largely absent from the W3C DPV instrument, and in the online service market, this gap reflects the absence of private records for sousveillance based transparency and trust, to enable scalable data governance controls.
- o These semantics are used to extend the EU DPR/GDPR policy and semantics for standard digital privacy transparency, for example,
  - ▪ Transparency by Design and Consent by Default, extends, EU DPR, Art, 27 Data Protection by Design and Default ,
  - ▪ Record of Notice Activity (RONA), and a Notified Records of processing activities(N-ROPA), extends EU DPR Art 31 RoPA for vector 1, personal data control governance

## Challenges for Implementing the Three Vectors of Data Governance

Data protection remains primarily a national or regional regulatory mechanism, typically focused on in-person, non-digital notice and consent requirements. As a result, most existing standards and privacy policy frameworks are not well-suited for co-regulation or for addressing online notice and consent. These governance frameworks tend to rely on contractual agreements and terms and conditions, without requiring demonstrable proof that notice has been given or understood. This approach exploits the transparency gap between analogue (physical) privacy contexts and digital environments, and fails to require evidence of informed consent.

### 2.2.2.1.         Regulatory Gaps in Digital Consent:

•       Recent legal decisions—such as those concerning Google's use of "necessary" cookies and the IAB (Interactive Advertising Bureau) Transparency and Consent Framework (TCF)—have found that these mechanisms do not constitute valid consent. Courts have ruled that the IAB acts as the PII (Personally Identifiable Information) Controller for cookies and is therefore liable for their use. Such rulings highlight that current digital consent infrastructures, including the use of cookies as digital receipts, are non-compliant.

•       Permission Fatigue and Dark Patterns:  The phenomenon often described as "consent fatigue" is more accurately characterized as "permission fatigue." This is a consumer protection issue, where security-based dark patterns leverage misinformation to undermine genuine consent. Terms and conditions are typically presented after user identification, offering permissions without real choice or control over tracking. There is usually no record or legal proof that individuals have read or understood the terms, nor is there evidence they comprehend the privacy risks or impacts. The prevailing analogue privacy tools—such as opt-in tick boxes— are primarily designed to indicate agreement with contract-based privacy policies, which do not provide meaningful privacy rights or controls in online contexts. These mechanisms can be altered at any time, often without notice, allowing for secondary use of data without transparency, consent, or permission.

• Lack of Enforced Transparency in Online Services: Online platforms, including major players like Google, often operate without enforced or standardized transparency. Self-regulation and the use of misinformation enable these services to secretly identify, collect, and process personal data. They create user identifiers and store them on devices without explicit permission, tracking individuals across borders for commercial gain. This practice undermines the security and privacy rights of non-Americans by circumventing privacy protections in other jurisdictions and bypassing the transparency required for valid consent. For example, in the United States, the ability to opt out of tracking is framed as a consumer protection measure, but only after digital privacy has already been compromised.

### 2.2.3. Updating the use of Normative Standards for Transparency Modalities and Digital conformance to consent.

The OPN Glassbox framework relies up open and free to access international standards that are suitable for implementing a standard operational privacy notice, that produces a notified record of processing activity, (similar conceptually to two factor authentication, this provides a 3 factor auth flow) in accordance with and specified by Convention 108+Treaty. Guided by the best operational governance practices in the EU Data Protection Regulation (DPR)

### 2.2.3.1. ISO/IEC JTC 1 WG 5 29100:2024

ISO/IEC 29100 is the international standard that has evolved and is interoperable with Convention 108 and 108+, it provides a privacy framework that specifies a common technical privacy terminology that can be used / mapped to privacy terminology in regulations, as it defines the actors and their roles in processing personally identifiable information (PII); describes privacy safeguarding considerations; and provides references to known privacy principles for information technology. Which are further refined for use in the OPN-Glassbox Model.

The OPN Glassbox-model improves upon the Privacy Principles my modifying the order in accordance with their implementation importance, order and priority in accordance with Article 29/EDPB Guidance on Transparency (see appendix)

The principles are represented here to modernise these data protection and data controller focused privacy principles to prioritise transparency, especially as the first requirement for identification, location tracking, and profiling surveillance.

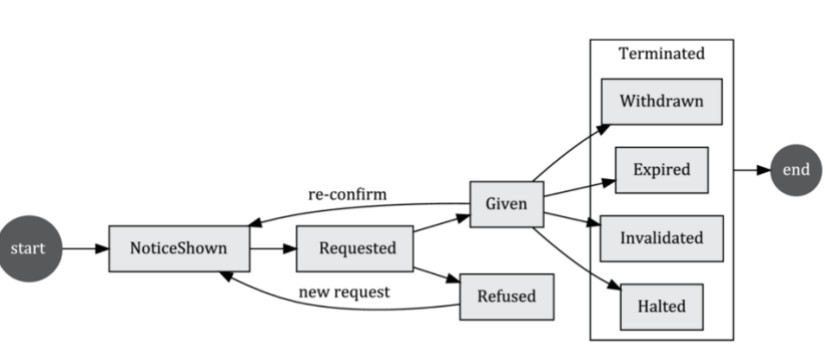

| Table 3 – The privacy principles of ISO/IEC 29100 | 0PN Glassbox Data Governance |
|---|---|
| 1. Consent and choice | **Prioritised According to Implementation** |
| 2. Purpose legitimacy and specification | 7. Openness, transparency and notice |
| 3. Collection limitation | 9. Accountability / Authority |
| 4. Data minimization | 1. Consent and choice |
| 5. Use, retention and disclosure limitation | 2. Purpose legitimacy and specification |
| 6. Accuracy and quality | 3. Collection limitation |
| 7. Openness, transparency and notice | 4. Data minimization |
| 8. Individual participation and access | 5. Use, retention and disclosure limitation |
| 9. Accountability | 6. Accuracy and quality |
| 10. Information security | 8. Individual participation and access |
| 11. Privacy compliance | 10. Information security |
| | 11. Privacy compliance |

**Figure 1:** ISO/IEC 29100:2024 (Information technology — Security techniques — Privacy framework) (https://www.iso.org/standard/80392.html).

### 2.2.3.2.    ISO/IEC 29184 Online privacy notice and consent

This standard is the companion document for the consent record information structure, drafted prior to the GDPR, and the EU GPR, which introduces Operational Personal Data, or lawful access privacy, which supersede 29184, Although, while it is out of date, it should be reviewed and updated to include lawful access and co-regulation.

### 2.2.3.3.    27560: Technical Specification,   Consent record information structure

Currently, this standard is defined as a data protection model, where the PII Principle is already identified or linked to a User-ID. This is evident as the user-id is in the header of the consent record,  without any proof of notice and consent for this identifier.   Evidence in and of itself that identification, (personal data collection) has occurred before this consent record is generated. This is typical of 'data protection' brick and mortar privacy regulation.

**ISO/IEC DTS 27560:2023(E)**

Figure B.1 — Overview of a typical consent record life cycle

**Figure   2:**   ISO/IEC   27560:2023   Consent   Record   Information   Structure. (https://www.iso.org/standard/80392.html). Consent lifecycle is defined for Data Controller,

- Diagram from 27560:2023 TS, representing the permission flow for consent in the context of of data protection regulation.  Which is written for in person, analogue privacy legislation,

wherein the notice, consent and permission are dealt with in a physical or person to person context without digital service intermediary, and self- regulated.

- 27560:2023 TS, also introduces significant security risk by defining a new role called a Party, without the ISO/IEC 29100 PII Controller and Processors distinction, presenting a schema that uses a "party id" to  back door access to personal data, for (unregulated stakeholders).

**Figure 2:** ISO/IEC 27560:2023 Consent Record Information Structure. Information structure schema, (https://www.iso.org/standard/80392.html).

In 27560:2023, the PII Principal is pre-identified, and there is no record of the permission, consent record, no proof of consent, and therefore not capable of producing a valid consent record for digital identification technology, which was the design and purpose of the MVCR specification upon which 27560 is based.[3]

While this schema might comply with the California Consumer Privacy Act, which mandates an 'opt-out' of tracking, meaning its is not only surveillance (not privacy) by default, but it misinforms on its secretive placement of  digital receipts with identifiers on devices, allowing for 'cookie surveillance'.  Identification that happens out of scope of the regulation, as it is for "consumers" who are already identified as a consumer in a service context.

For non-US citizens, regardless of this law, personal information and communications are secretly kept by state and national security services, limiting the privacy preserving security benefits to US citizens only.[4]

This is not complaint Quebec privacy law,[5] represents a significant national and economic security issue for Canadians, as well as the EU, and other commonwealth countries.[6]

---

[3] Annex A: Table of the Legal references invalidating 27560:2023 TS: Consent Record information structure,

[4] Referring to the USA, Foreign Surveillance Intelligence Act, section 802

[5] ACT RESPECTING THE PROTECTION OF PERSONAL INFORMATION IN THE PRIVATE
SECTOR, Section 8.1 any person who
collects personal information from the person concerned using technology that includes functions allowing
the person concerned to be identified, located or profiled must first inform the person
(1) of the use of such technology; and
(2) of the means available to activate the functions that allow a person to be identified, located or "Profiling" means the
collection and use of personal information to assess certain characteristics of a natural person, in particular for the
purpose of analyzing that person's work performance, economic situation, health, personal preferences, interests or
behaviour.

[6] Note: at the time of writing this paper, the governance of the digital borders between Canada and the USA is actively being challenged by the Trump administration, which has threatened to make Canada the 51st state, claiming Canada to already to be the 51st digital state.

Identified, tracked, located, profiled, requires not only explicit permission but also the control of this feature to be held by the individual unless delegated with a record of notice and consent to prove it.

*Indicating that 27560 does not produce valid consent record information structure and is not compliant with Convention 108+ or Canada's high bar for permission and consent.*

Surveillance by default is culturally impolite for many reasons. Socially and culturally in Canada, its impolite, to use track and process information, before asking permission and confirming the consent of the individual. Legally, this form of permission does not come with valid consent. Presenting significant Security Risks

- National security and personal data sovereignty
    - Canadian digital sovereignty and the capacity to govern technologies that integrate "User-Id" identification and tracking systems internationally without transparency.
- Environmental Security impacts,
    - The use of non-regulated transparency and consent records like cookies, hijack the digital receipt, and auction this for advertising profits. Rather then, providing this receipt to the individual, so they can use it to track the impact digital choices and relationships have on the environment. In this trade-off, both the Controller and the Principal avoid accountability for the their actions online. Enabling mass misinformation, division, and lack of consensus.

### 2.2.4. Non-Normative

W3C Data Privacy Vocabulary, is strictly defined to the GDPR, and for Data Controllers which greatly limits its utility for notice semantics, but is unparallelled in its utility to make standard machine readable notice records and receipts, which might even be human readable, but without a standard for transparency and consent semantics, not meaning for the control of data with consent. [7]

- Missing Co-Regulator / Consent Semantics in DPV
    - Humans manage consent, systems manage permissions
    - Humans self-assert their identity which requires transparency to be valid (sousveillance), while system assert an identifier for identification (surveillance)
    - A notice receipt, which is returned with permissions and preferences indicated is a record of consent, that can be held by a Controller and reported upon,
    - In Data Control with Consent, Consent is sovereign, the service is the 'user' of personal information or an individuals identity,
    - In Consent, the purpose is picked and asserted by the individual engaging and providing permission, this purpose should be clarified in notice prior to identification, data processing, or data collection (which is co-regulated in this model)

---

[7] Culturally, in Canada, its not only polite to ask for permission to identify an individual or introduce a new purpose, its the law, a notice of who the controller is and the governance framework they operate is required for consent to identification and digital surveillance. Which are rules not required in the USA.

- A notice by its nature, must be read and understood with consent and consensus of the individual for its comprehension. Meaning in the context of transparency semantics, regardless of the legal justification, the first notice is consent by default. (and requires enough trust for the individual to read in the first place)

## 3. Meta-Digital Privacy Model: 0PN Glass-box data governance notice record structure

Glassbox Data Governance Model mirrors other identification based record and receipt systems, that require a registry and event log but with a significant difference in that the registry, record, receipt, notice, notification and disclosure system is managed as public infrastructure, in accordance with or compared to the  standard adequate, privacy framework.

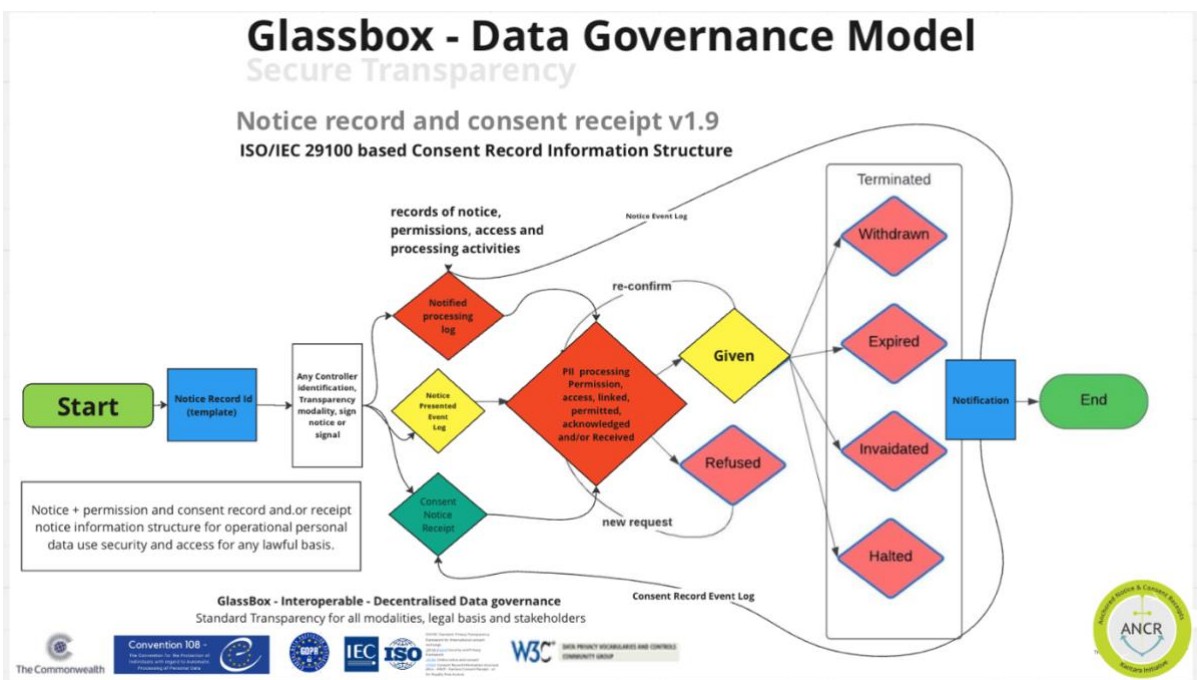

Figure 3 0PN Glassbox Data Governance record information structure contribution to the Standards Council of Canada Data Governance Collaborative

This diagram presents a notice record and receipt model for the generation of notified record of processing activities, referred to generally in this meta model as the PII Controller notice identification record,  which  can be utilized for 3 vectors of data governance.

Vector 1. Personal Data Control – the first red triangle generates a record for an legal justification, can be used for private data governance, for employee based governance, and industrial contexts whre PII is not used at all.
Vector 2: Data Protection and Controller based data governance – the first yellow triangle  - depicts the flow that is currently in which require and operational privacy (or policy) notice format and the presentation of the Controller identity, and responsibilities of the user.
Vector 3: depicts a record and receipt exchange flow, which generates a consent record with proof of notice and the text provided in the record.

# 4. OPN, Digital Privacy, Security and Consent Considerations

The OPN, operational privacy notice is a privacy model pattern the uses privacy regulation to govern the use of identification management. Utilizing the OPN- (Canadian-International) Notice record and receipt structure and schema for the 27560 consent record information technical specification address these issues, and complete the original MVCR works.

OPN, for Glassbox data governance provides security and operational transparency capabilities for all vectors of governance describe. It provides for a co-regulatory transparency model, which enables individuals to manage their own security, without requiring what is referred to as "Digital Identification Trust". It provides secure online transparency, addressing major security and privacy risks by using a PII Controller identifier in the record header, and a notice record and consent receipt information structure schema, where all stakeholders are defined as a PII Controller, with additional defined roles, of processor, joint-controller, etc. No (back-door) unregulated 3rd parties.

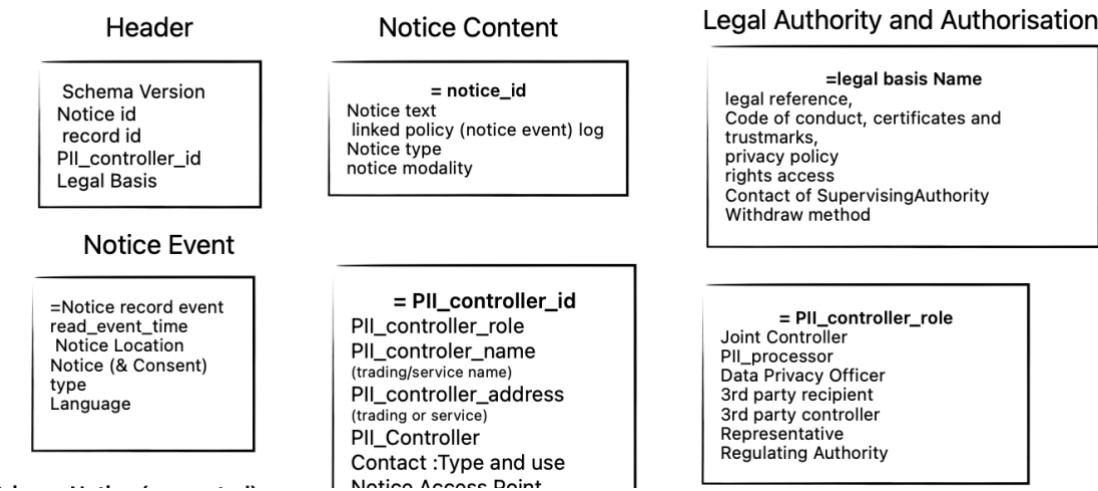

**Figure 4:** 27560 Canadian International compliant Schema profile, for Consent Record Information Structure. Information structure schema, (https://0pn.org).

## 4.1. Secure and Operational Transparency

The Operational Privacy (or Policy) Notice standard pattern wherein a notice, along with the Controller Identity, is presented to the PII Principal, and once the PII Principal interacts with it, a record of this proof notice, is generated and provided as a notice receipt to the principle.

This operational transparency pattern is secure because the controller identification is used to generate the receipt, and no PII is provided or permissioned until the receipt is then returned and through this action consented.

"where transparency is required for Consent or another legitimate basis define by law" all other legal basis," as well as in line with GDPR Guidance,
While in the EU DPR which mirrors the GDPR,

"The concept of transparency in the GDPR is user-centric rather than legalistic" as is required for "securing every individual's right to protection of his or her personal data" (Article 3.1-Convention 108+).

"64. Furthermore, there may be situations where a data controller is processing personal data that does not require the identification of a data subject (for example, with pseudonymized data). In such cases, Article 11.1 may also be relevant as it states that a data controller shall not be obliged to maintain, acquire, or process additional information to identify the data subject solely to comply with the GDPR."[8]

Secure transparency, wherein a standard presentation of a Controller's identity, and this control-id is used to generate a notice record, (instead of a User-ID), presents and architecture that mirrors, the user-id, and uses standard transparency to reduce the burden of notice for the Controller.

A notice receipt for the Controller identity presentation is required in order to engage with any online service provider. The Controller, the individual first generates this anchor receipt, this receipt, is then utilized privately on the individual's device, sharing from secure private digital space, eliminating the risks of providing PII using a un-regulated website.

OPN, Secure digital transparency, operationalizes analogue privacy notice and practices, through the principal of transparency.

prioritizing the transparency principle reduces even eliminates privacy risks inherent to analogue privacy principles, namely; 3. Collection limitation, 4. Data minimization, 5. Use, retention and disclosure limitation existing identification management risks;[9]

## 4.2.    Co-Regulatory Semantics

In co-regulatory frame, Trusted security, and the semantics of trust are relative to the stakeholder with transparency and control of the data.

- In vector 1, personal data control removes/reduces the requirement of data protection and trust
- In vector 2, data protection makes the trust decisions for people
- In vector 3, co-regulated online transparency rules are standard so that transparency is provided according to the legal justification, and when not the individual, and Privacy Officer or Privacy Regulator, holds that transparency as a fiduciary responsibility,
- in all cases, whether co-regulatory or not processing practices can be measured against the international transparency adequacy standard and baseline.
-

### 4.2.1.    Data Control so Data Protection is not needed

OPN, is the commons (public-data) digital privacy model pattern,[10] it opens up the digital privacy technical architecture, enabling innovation in the use of notice receipts for decentralized consent-based authorization, without having to be digitally identified online. This is the opposite of what currently happens) utilizing a well-known pattern called ZKP (Zero Knowledge Proof) authentication. In this way the individual can remain anonymous, but assured online, only identified or profiled, when required by a service with the authority to do so. No longer a requirement to provide or publish PII

---

8 Referencing Recital 64, EDPB, Guidelines 05/2020 on consent under Regulation 2016/679

9 ; ISO/IEC 29100– Table 3 Privacy Principals, when operationalised by the transparency principle, using the OPN – Privacy model pattern, eliminates / reduces the need for analogue privacy principles.

10 3.3

model pattern,

general, reusable model (3.2) or model part that can be used as a solution to a commonly occurring problem within a given context in system or software design

[SOURCE: ISO/IEC/IEEE 24641:2023, 3.1.21]

or personal data on or over the internet, over and over again, for each service provider. (dramatically decreasing privacy risks, and the risks of a data breach)

### 4.2.1.1.    Comparative Reference

Much like mid evil times when transferring gold in a cart from a bank in Florence to a bank in Paris was replaced with a notarized bank receipt. The operational privacy notice pattern removes the initial requirement for PII to be transferred across the internet to start with.

0PN Eliminates the initial risk vector of PII being captured and linked to a user-id, without permission, or consent, by requiring macro-data, system generated identifiers to be made transparent at the point of controller identification, before capture and use as meta-data.

For example,

This OPN pattern is implemented by Generating a Notice record and twinned notice receipt using the control-id, saving this to a secure and personal device. A notice receipt becomes a consent notice receipt or consent token once it is sent back to the PII controller.

The action of sending the receipt back standardized the permissions and can include, another notice receipt, which can provide access to data that is held by 3rd Party Data Controlllers.

The attachment of cosent notice rececipts, turns the consent receipt into a remote access consent token[11].

In this way, the Convention 108_ standard operational privacy notice flow provides the means to a) secure the notice and notification from being tracked, b) enable the private-personal sharing and administration of personal data, mitigating the risks of data scraping, key-logging and cookie surveillance.

For example,

- Converting cookies (digital surveillance receipts) into standard notice (digital sousveillance receipts), provides the individual with the capacity to keep a personal (trust capable) clean data record of the digital relationships, and to keep this private record as an anchored notice and consent record, to update, modify and control personal data and its access after its has been provided, independently, and autonomously from the Controller.
- a direct relationship with the controller, cutting out unregulated intermediaries and data brokers, and addressing national data security and sovereignty risks.

### 4.3.  Decentralized Data Governance, Interoperability and Enforcement

The evolution of Glassbox data governance is again very similar to the evolution and development of banking governance and regulation. Standard notice record and receipts have the same impact of adding receipts to cash-registers, regulating the robber barons, had on the market in the early 1800's. Wherein a receipt could be used to return a goods or geta refund for a service, decentralizing the governance required in the use of currency. Enabling the individual to manage their own dispute and return goods.

Similarly, this standard notice records and consent notice receipts, for Convention 108+ utilized as the international privacy standard deploys transparency as a tool for human regulated interoperability, enabling the individual to use the legal basis of consent to manage and withdraw permissions with the authority of consent, in a centralized manner, in context rather than per service,

---

[11] Consent Token, is a notice receipt that is returned to the PII Controller, with data portability, assurances, currency and data control requirements added to the receipt. AI Governance, consent token can contain multiple notice receipts and tokens, enabling the automation of private and personal intelligence.

using analogue privacy controls out of digital context. In a very similar manner conceptually to that of a transaction receipt, the consented notice receipt, decentralizes data governance, enabling interoperability and co-regulation.

### 4.3.1.      International and Multi-Lateral Transparency Governance to Govern Surveillance Capitalism

The Global Challenge of regulating the use of surveillance capitalism, backed by state policy, can be addressed to plug the secret surveillance holes with the OPN- Glassbox transparency, utilising the model pattern.

This can be demonstrated with  ANCR Transparency Performance Indicator Report (TPI-R) from the Kantara Initiative, which is currently being trialed in Quebec, Canada for regulatory enorcement innovation,

ANCR Transparency Performance Indicator Report (TPI-R)

This, specification, expected to be published in July, implements and reports on the 4 transparency performance indicators, that measure the performance of 'notice', (broadly referring to any notice, notification or disclosure, or surveillance signal), to determine if consent, or any other legitimate basis valid for the use of digital identification.

The Quebec trial of the  TPI-R involves the submission of 3 types of transparency enforcement complaint to the regulator, where the TPI-R is used as evidence as to the performance of consent.

This conformance and compliance assessment is provided referencing Quebec Privacy Law that came into force in 2023.

The initial complaint is about a medical clinic website, which uses cookies and key-logging even though cookies are turned off, targeting the 'Necessary Cookie' Fraud.

The second complaint, utilizes the TPI-R to report on the compliance of 'Necessary Cookie' as a collective complaint, with 28 participants, across an array of regulatory considerations.  Providing the use case for a single TPI-R to be resisted by many, which includes a private right of action for upt to 50k per complaint

The third complaint utilizes the TPI-R, to provide evidence that the Chrome Browser 1is not compliant but instead deceptive, in that it automatically identifies and profiles individuals without Controller Notice, or Purpose Transparency.   The maximum administrative fine is $25 million, or 4% of worldwide turn-over.

 These  complaints  were  launched  on  July  1th, 2025 and uses the TPI-R to assess if the use of identification technologies in the Chrome Browsers constitutes legitimate transparency and valid consent against

This trail of the TPI-Report aims to test the path for automating transparency enforcement with a standard conformance and compliance tool  but, also, the use of  the same complaint that references not only Quebec Law 25, but also Convention 108+ and the GDPR. (See Appendix 7.3)

This provides the framework for a multi--lateral international surveillance capitalism governance enforcement  capacity.  Where  a    complaint  can  be  enforced  in  multiple  jurisdictions  at  once.

Combined  these  conformance  and  compliance  measures  provide  for  the  governance  of   big technology companies who have profited from the lack of international privacy transparency

standard, and capacity for international multi-lateral transparency market with effective regulatory enforcement.   To find out more checkout – Globalprivacyrights.org

## 4.4.     Operational Personal Data for Real Time Lawful Access and Consent

Glassbox data governance architecture provides for front door, real time access, with the generic record and receipt components,

- Notice Registry,
- Notice event log,
- Controller Notice identity, and
- The OPN privacy model, public data notary, and transparency registry,  which provides for the operationalisation of personal data for any legal justification, with 4 levels of transparency assurance (LoTA).

LoTA 1 – Low assurance but uses public notice record  standard to self-assert transparency, providing slightly higher assurance than current market dark patterns.
LoTA 2 – Controller provided receipt, unregistered or notarised, peer to peer,
LoTA 3 – Controller registered data notary, $3^{rd}$ party notary of records and receipts
LoTA 3 – Real time monitoring and access, with dynamic transparency, and realtime access for legitimate purpose, with the authority, all access and processing is logged, in a data notary ledger, which is secured according to legal requirements.

### 4.4.1.     Anonymous, secret and pseudonymous levels of surveillance assurance

Digital Transparency Index - Governance Registration and Interoperability

> Very much modelled off of the ground breaking (at its time) 1998 Data Controller Registry in the UK and the SORN, System of Recorded Notices  in the USA. A controller transparency registry or index is administered as public digital infrastructure, able to validate, verify, and monitor notified records of processing, records of notice activity, and the required public records of notice for international data governance.
> Providing services for the PII Principle which enable data portability, access, control and use transparency, like a bank account does for currency,

## 5.  Conclusion

The digital era demands a fundamental transformation in how privacy, transparency, and consent are governed. This paper has introduced the OPN Glassbox Data Governance meta-model which uses the authoritative CoE Convention 108+ for standard digital privacy transparency.
This model introduces governance for 3 vectors of data governance to address analog shortcoming of data protection regulation,  which are ill-suited for the complexities of the digital environment. This model leverages internationally recognized standards such as Convention 108+, ISO/IEC 29100, and the W3C Data Privacy Vocabulary, the Glassbox model establishes a robust, interoperable

structure for operational privacy notices, consent receipts, and real-time lawful access.

The Glassbox approach reframes privacy governance by:
• Standardizing transparency and consent across all vectors of data control, ensuring that individuals, organizations, and regulators operate with a shared, verifiable information structure.
• Empowering individuals to manage and audit their personal data relationships through receipt-based records, thereby restoring autonomy and trust in digital interactions.
• Supporting co-regulation and cross-jurisdictional enforcement by providing a public infrastructure for notice, consent, and dispute resolution, which is essential for addressing the global nature of digital surveillance and data flows.
• Enhancing security, sovereignty, and environmental accountability by making the impacts and permissions of digital actions transparent and auditable for all stakeholders.

In sum operationalizing transparency and consent with technical standards—rather than mere policy aspirations—the Glassbox model closes critical gaps in current data protection regimes to defend against foreign surveillance. It provides a scalable path for regulators, service providers, and individuals to achieve compliance, foster innovation, and uphold fundamental rights in the digital commons.

The adoption of this meta-privacy transparency model is not just a digital privacy upgrade; it is a necessary evolution for a trustworthy, inclusive, and resilient digital society. As international standards like Convention 108+ and Quebec Law 25 come into force, the Glassbox framework offers a practical blueprint for implementing digital privacy transparency that is both globally interoperable and locally enforceable. This shift from opaque, contract-based governance to transparent, receipt-based accountability is essential for securing personal data, strengthening regulatory oversight, and enabling individuals to meaningfully participate in the digital world.

## 6. Citations and bibliographies

• Article 29 Working Party, "Guidelines on Transparency under Regulation 2016/679"
• Council of Europe Convention 108+ (2025)
• EU General Data Protection Regulation (GDPR)
• EU Data Protection Regulation (DPR)
• Quebec Law 25 (2023)
• ISO/IEC 29100:2024
• ISO/IEC 29184:2020
• ISO/IEC 27560:2023

- ISO/iEC 27564: Privacy protection — Guidance on the use of models for privacy engineering
- ISO/IEC 27091 (Working Draft)
- W3C Data Privacy Vocabulary (DPV)
- ISO/IEC (Proposed)  Canadian International Notice Record and Receipt information structure profile
- California Consumer Privacy Act, California Civil Code Section 1798.100, enacted January 1, 2020. Available at: California Department of Justice
- Kantara Initiative ANCR Transparency Performance Indicator Report (TPI-R), https://kantara.atlassian.net/wiki/spaces/WA/blog/2025/02/14/875200525/ANCR+WG+Introduces+Transparency+Performance+Indicator+Benchmark+for+Valid+Consent+to+Identification
- European Data Protection Board. (2020). Guidelines 05/2020 on consent under Regulation 2016/679. Retrieved from https://edpb.europa.eu/sites/default/files/files/file1/edpb_guidelines_202005_consent_en.pdf
- OECD. (2025). Enhancing Access to and Sharing of Data in the Age of Artificial Intelligence. Retrieved from www.oecd.org

## 7. Appendices

Appendices should be added after the references. Note that in the appendix, sections are lettered, not numbered.

### 7.1. Article 29 Working Party, GDPR Guidance on Transparency

| 1 | Transparency is an overarching obligation under the GDPR applying to three central areas: as compliance with transparency is required in relation to data processing under Directive (EU) 2016/6803, these guidelines also apply to the interpretation of that principle. |
|---|---|
| 2 | in addition to the requirements that data must be processed lawfully and fairly, **transparency is now included as a fundamental aspect of these principles.** |
| 4. | The concept of transparency in the GDPR is user-centric rather than legalistic |
| 5 | The transparency requirements in the GDPR apply irrespective of the legal basis for processing and throughout the life cycle of processing |

### 7.2. EU GPR – Operational Personal Data Requirements

| Article | Title | Description |
|---|---|---|
| 70 | Scope of the Chapter | Applies to processing of operational personal data by Union bodies, offices, and agencies when carrying out activities under Chapter 4 or 5 of Title V of Part Three TFEU. |

| Article | Title | Description |
|---|---|---|
| 71 | Principles Relating to Processing of Operational Personal Data | Lawfulness and fairness, Purpose limitation, Data minimisation, Accuracy, Storage limitation, Integrity and confidentiality, Accountability. |
| 72 | Lawfulness of Processing of Operational Personal Data | Processing is lawful only if necessary for the performance of a task by Union bodies under Union law. |
| 73 | Distinction Between Different Categories of Data Subjects | Controllers must distinguish, as far as possible, between categories of data subjects (e.g., suspects, victims, witnesses). |
| 74 | Distinction Between Operational Personal Data and Verification of Quality | Distinguish data based on facts from those based on personal assessments. Ensure data quality before transmission. |
| 75 | Specific Processing Conditions | Inform recipients of specific processing conditions. Comply with conditions set by transmitting authorities. |
| 76 | Processing of Special Categories of Operational Personal Data | Processing of sensitive data (e.g., racial origin, health) allowed only when strictly necessary and with safeguards. |
| 77 | Automated Individual Decision-Making, Including Profiling | Prohibits decisions based solely on automated processing unless authorised by law and with safeguards. |
| 78 | Communication and Modalities for Exercising Data Subject Rights | Controllers must provide information in a concise, intelligible, and accessible form. Facilitate exercise of rights. |
| 79 | Information to be Made Available or Given to the Data Subject | Controllers must provide information on identity, purposes, rights, and recipients. |
| 80 | Right of Access by the Data Subject | Data subjects have the right to access their operational personal data and related information. |
| 81 | Limitations to the Right of Access | Access may be restricted to protect investigations, public security, or rights of others. |
| 82 | Right to Rectification or Erasure and Restriction of Processing | Data subjects can request rectification or erasure of inaccurate or unlawfully processed data. |
| 83 | Right of Access in Criminal Investigations and Proceedings | Access to data originating from competent authorities is subject to consultation with those authorities. |
| 84 | Exercise of Rights by the Data Subject and Verification by the Supervisor | Data subjects may exercise rights through the European Data Protection Supervisor (EDPS). |
| 85 | Data Protection by Design and by Default | Controllers must implement technical and organisational measures to ensure data protection principles. |
| 86 | Joint Controllers | Joint controllers must transparently determine their respective responsibilities. |

| Article | Title | Description |
|---|---|---|
| 87 | Processor | Processors must provide sufficient guarantees and act only on controller instructions. |
| 88 | Logging | Controllers must keep logs of processing operations for verification and security. |
| 89 | Data Protection Impact Assessment | Required for processing likely to result in high risk to data subjects' rights. |
| 90 | Prior Consultation of the European Data Protection Supervisor | Controllers must consult the EDPS before high-risk processing. |
| 91 | Security of Processing of Operational Personal Data | Controllers and processors must implement appropriate security measures. |
| 92 | Notification of a Personal Data Breach to the Supervisor | Controllers must notify the EDPS of personal data breaches within 72 hours. |
| 93 | Communication of a Personal Data Breach to the Data Subject | Data subjects must be informed of breaches likely to result in high risk. |
| 94 | Transfer of Operational Personal Data to Third Countries and International Organisations | Transfers allowed only under strict conditions ensuring adequate protection. |

## 7.3. Multi-Lateral Transparency Enforcement and Interoperability: Legal Reference For International Transparency and Consent Enforcement

| Results | Risk Rating | Measures | Quebec Law 65 | CAI Guidance | ISO/IEC 29100 | GDPR | Convention 108+ |
|---|---|---|---|---|---|---|---|
| TPI 1: | 1 | timing of controller identification conformant and compliant with Controller identification requirement before disclosing PII | B.3 Consent and Collection Comply with its **obligation of transparency** by providing accurate and complete information to the persons concerned when the collection is made from them4. | CAI (pg6) B.9. Timing of Consent An organization must obtain consent before performing the actions to which it relates. | 6.2 Consent & Choice

• providing PII principals, before obtaining consent, with the information indicated by the openness, | Article 13.1 b), and 141, a) and b) all data are obtained, provide the data subject with all of the following information: (a) the identity and the contact details of | Recital 68, p.23 68. Certain essential information has to be compulsorily provided in a proactive manner by the controller to the data subjects when directly or indirectly (not |

### 7.3. Multi-Lateral Transparency Enforcement and Interoperability: Legal Reference For International Transparency and Consent Enforcement

| Results | Risk Rating | Measures | Quebec Law 65 | CAI Guidance | ISO/IEC 29100 | GDPR | Convention 108+ |
|---|---|---|---|---|---|---|---|
| | | | | | notice and choice principle | the controller;( b) the contact details of the data protection officer. | through the data subject but through a third-party) collecting their data, subject to the possibility to provide for exceptions. |
| TPI 2 0 | | identification provided, inaccurate, and or not registered to operate (e.g. a medical clinic at that address) in Quebec | B.3 Consent and Collection Comply with its **obligation of transparency** by providing accurate and complete information to the persons concerned when the collection is made from them4. | | 5.6 pg.13 An external privacy policy provides outsiders to the organization with a notice of the organization's privacy practices, as well as other relevant information such as the identity and official address of the PII controller, contact points from which PII principals can obtain additional information, etc. In the context of this framework, the term | Article 13.1, 14.1 (a) the identity and the contact details of the controller and, where applicable, of the controller's representative; (b) the contact details of the data protection officer, where applicable; | Recital 68 p.23 Information on the name and address of the controller – the right of everyone not to be subject to a purely automated decision significantly affecting them without having their views taken into consideration (littera a.) ; – the right of everyone to request |

### 7.3. Multi-Lateral Transparency Enforcement and Interoperability: Legal Reference For International Transparency and Consent Enforcement

| Results | Risk Rating | Measures | Quebec Law 65 | CAI Guidance | ISO/IEC 29100 | GDPR | Convention 108+ |
|---|---|---|---|---|---|---|---|
| | | | | | "privacy policy" is used to refer to the internal privacy policy of an organization. External privacy policies are referred to as **notices**. | | confirmation of a processing of data relating to them and (or co-controllers), the legal basis and the purposes of the data processing, the categories of data processed to access the data at reasonable intervals and without excessive delay or expense (littera b.); and recipients, as well as the means of exercising the – the right of everyone to be provided, on rights can be provided in |

### 7.3. Multi-Lateral Transparency Enforcement and Interoperability: Legal Reference For International Transparency and Consent Enforcement

| Results | Risk Rating | Measures | Quebec Law 65 | CAI Guidance | ISO/IEC 29100 | GDPR | Convention 108+ |
|---|---|---|---|---|---|---|---|
| | | | | | | | any appropriate format |
| TPI 3 -1 | | Data privacy access point is not easily accessed, is not operational | 8.1. In addition to the information that must be provided in accordance with section 8, any person who collects personal information from the person concerned using technology that includes functions allowing the person concerned to be **identified** ,located or profiled must first inform the person (1) of the use of such technology; and (2) of the means available to activate the functions that allow a person to be identified, located or profiled. | B.2 Methods of Through rights (access, rectification, etc.) or remedies 25/CAI Guidance ISO/IEC 29100 Control a) (complaint to an organization or the CAI, etc.). To ensure that individuals can exercise these rights in full knowledge of the facts, the laws provide for **transpar ency** obligations for organizations ; | **6.9 Individual participation and access (pg.17)** Adhering to the individual participation and access principle means: - giving PII principals the ability to access and review their PII, provided their identity is first authenticated with an appropriate level of assurance and such access is not prohibited by applicable law; | 13.1 (b), 14.1 (b) rights access | "Article 8 - Transparency of processing 68. can be provided in any appropriate format (either through a website, technological tools on per-sonal devices, etc.) as long as the information is fairly and effectively presented to the data subject. The information presented should be easily accessible, legible, understandable and adapted to the relevant data subjects (for example, |

## 7.3. Multi-Lateral Transparency Enforcement and Interoperability: Legal Reference For International Transparency and Consent Enforcement

| Results | Risk Rating | Measures | Quebec Law 65 | CAI Guidance | ISO/IEC 29100 | GDPR | Convention 108+ |
|---|---|---|---|---|---|---|---|
| | | | | | | | in a child friendly language where necessary). Any additional information that is necessary to ensure fair data processing." |
| TPI 4 | -1 | Sensitive Health Information is collected, processed and transferred internationally with out notice, permission, or the legal authority of consent. Breaking security and causing secret data breaches | Law 25 - 110 s12. (3)\| Law 25 – 144 "(6) the other measures taken to ensure the confidentiality and security of personal information in accordance with this Act."; Law 25 v-159(4) does not take the security measures necessary to ensure the protection of the personal information in accordance with section 10; | 17. Every person carrying on an enterprise in Québec who communicates personal information outside Québec or entrusts a person outside Québec with the task of holding, using or communicating such information on his behalf must first take all reasonable steps to ensure (1) that the information will not be | 6.11 Information security Adhering to the information security principle means: Implementing controls in proportion to the likelihood and severity of the potential consequences, the sensitivity of the PII, the number of PII principals that might be affected, and the context in which it is held; - limiting | Recital 39 … Personal data should be processed in a manner that ensures appropriate security and confidentiality of the personal data, including for preventing unauthorised access to or use of personal data and the equipment used for the processing. | Article 7 - Data Security 63 p.22 & 110. pg. 28 63. Security measures should take into account the current state of the art of data-security methods and techniques in the field of data processing. Their cost should be commensurate with the seriousness and probability of the |

## 7.3. Multi-Lateral Transparency Enforcement and Interoperability: Legal Reference For International Transparency and Consent Enforcement

| Results | Risk Rating | Measures | Quebec Law 65 | CAI Guidance | ISO/IEC 29100 | GDPR | Convention 108+ |
|---------|-------------|----------|---------------|--------------|---------------|------|-----------------|
| | | | | used for purposes not relevant to the object of the file or communicated to third persons without the consent of the persons concerned, except in cases similar to those described in sections 18 and 23; .... , the person must refuse to communicate the information or refuse to entrust a person or a body outside Québec with the task of holding, using or communicating it on behalf of the person carrying on the enterprise. | | | potential risks. Security measures should be kept under review and updated where necessary.

110. The level of protection should be assessed for each transfer or category of transfers. Various elements of the transfer should be examined such as: the type of data; the purposes and duration of processing for which the data are transferred; the respect of the rule of law by the country of final destination; the general and sectoral |

## 7.3. Multi-Lateral Transparency Enforcement and Interoperability: Legal Reference For International Transparency and Consent Enforcement

| Results | Risk Rating | Measures | Quebec Law 65 | CAI Guidance | ISO/IEC 29100 | GDPR | Convention 108+ |
|---|---|---|---|---|---|---|---|
| | | | | | | | legal rules applicable in the State or organisation in question; and the professional and security rules which apply there. |

## 7.4. Levels of Notarial Transparency Assurance for 3 Vectors of Data Governance

Each vector of governance can use the standard notice record and receipt, operational privacy notice standard pattern to provide a high level of assurance,

| Levels of Transparency Assurance | Description | MVCR v2 schema | Vectors 1. Personal Data Control | Vector 2. Data Protection | Vector 3. Co Regulation |
|---|---|---|---|---|---|
| Level 1- low risk disclosure Assurance * No Controller ID | Private, personal records of notice. | Self-asserted digital copy of notice | Local sign, for semi-private spaces – self asserted physical policy | Localised (video) or CCTV – Surveillance | UK ICO Controller Registry, |
| Level 2 * Controller ID | Peer to Peer, Controller is provided | Uses standard records | Keeps a record /receipt event log | Keeps a privacy policy for protection actions | ,is audited like health and safety inspections |
| Level 3 * Open Public Social Controller Notice and Idenity Profile | Public Data Notar, | Standard Records and Consent Receipts, re-used as consent tokens for secondary | Self generated receipts signed by controller | Controller generated receipts signed by the Principal | Notary Signed and stored records, with receipts provided to controller |

| | | purpose of use – PII transferred directly (cutting out intermediarie s) | | | | and principal |
|---|---|---|---|---|---|---|
| 8. Dynamic Transpare nt Data Governan ce | Dynamic Signalling | Independent and synchroni records of transparency, no direct linkable contact | Dynamic Transparen cy Signalling | real time anonymous access and regulated (data gov- notary) with pseudonymo us access controls | Banking Data – syest of record transparenc y, Glassbox currency governance . | |

## Acknowledgements

Thank you to Kantara Initiative, the CISWG and ANCR WG's, who put in all the volunteer time to support this notice profile. This work completes the Minimum Viable Consent Receipt (MVCR), which is what 27560 was originally drafted from.

The 0PN-Glassbox Framework is made available under by Global Privacy Rights, under a modified Royalty-Free Rand license, that can be used for operational privacy notice infrastructure and its policy as defined.

while universal digital transparency is touted here as the magic technical solution, it exists and is the result of many generations of human centric, social-technical economic evolution, deeply embedded in human cultures and society.

## Declaration on Generative AI

*Either:*
The author(s) have not employed any Generative AI tools.

*Or (by using the activity taxonomy in ceur-ws.org/genai-tax.html):*
During the preparation of this work, the author used Grammarly, spelling and editing suggestions, the author(s) reviewed and edited the content as needed and take(s) full responsibility for the publication's content.

