# OpenReview forum: "Digital Privacy Transparency, a Meta Privacy Model for Glass-boxed Data Governance"
_SEMANTiCS.cc/2025/Workshop/NXDG — Submitted to NXDG 2025_

### Official Review · ~Anelia_Kurteva1 · 2025-07-21
**Proposal for a meta privacy model for glass-boxed data governance, which while interesting and important is not well presented in the paper.**

**Rating:** 3
**Confidence:** 4

**Review:**

The paper discusses a meta privacy model for glass-boxed data governance. This is a topic of important and (personal) interest. However, I found the content disorganised, difficult to follow and it is unclear what the contribution is. I suggest the authors improve the paper significantly and resubmit at a suitable venue.

Specific comments:
The abstract needs to be rewritten so that it clearly states the challenges at hand (the motivation) and the proposed solution in this work. The introduction starts quite abruptly and does not actually introduce the main challenge and work in the field. I could not find references in the paper (only naming legislations), which is a drawback.

From the introduction, it is unclear what the contribution is. Is this just an analysis and discussion or a proposal for a new framework? The content is a bit confusing to me.

Terminology should be defined when first mentioned, with references and used consistently.
Sections 1 and 2 can be combined in one shorter section.

Table 1 is missing a caption.

There are some issues with the section numbering and it seems unnecessary to have such a nested list of sections.

The formatting and language of the paper need significant improvement. For example, sometimes different fonts have been used and sentences are too long they lose meaning.

Missing reference to DPV.

Unclear methodology and research question(s). Unclear if evaluation can be done and how, the limitations of the work and if future work is planned.

There are two figures with the same caption (Figure 2).

All the content till section 3 should be summarised (already 8 pages overviewing existing frameworks). More focus should be put on the actual contribution of the work.

Figure 3 is not readable when printed. This is not a big issue but could be improved easily.

The conclusions section sounds more like content that should be in the paper (e.g. in section 3).

The citation list is not up to standard in terms of formatting. The authors should follow the provided template.

The role of the appendix is unclear. It should be better connected to the paper content (e.g. in specific sections). The 1st sentence in the appendix should be deleted as it is from the template.

The title of the paper is also ambiguous as well. It should better reflect the paper (e.g. if it is an analysis or proposes a new solution).

---

### Official Review · ~Michiel_Fierens1 · 2025-07-23
**Added value, but in need of further clarifications, sources, and a better structure**

**Rating:** 4
**Confidence:** 4

**Review:**

The topic is definitely worthwhile and suitable for the NXDG Workshop. The development and explanation of the meta-privacy model for glass-box data governance fits the theme perfectly.

However, several modifications are needed to make the paper acceptable for publication. Currently, the paper goes in different directions without clear delineation and structure and in-depth treatment of the issues.

(1) Several terms are introduced without providing any nuance and clarification. Likewise, strong statements are placed for which no argumentation is made. It feels as if the reader is making various assumptions without really taking the reader along in the precise explanation or reasoning behind them, which means that they are not necessarily verifiable for the reader.

- Nuances between EU data protection and US/Common Law traditions: how do they compare/link to each other, why is this relevant? Differences US and non-US citizens, why relevant?
- Nuances between personal data legislation/governance and non-personal data legislation/governance.
- Nuance when mentioning that GDPR and CCPA are merely updates to legislation based on older context?
- What is meant with peer to peer brick and mortar contexts? Please clarify analogue privacy context.
- Terms Controllers, Data Controller, PII Controller used interchangeably without further clarification.
- Why are cybersecurity vulnerabilities only mentioned in passing, and why?
- Further arguments needed for statement that governance frameworks tend to rely on contractual agreements and terms and conditions.
- Additional sources and explanation needed for the regulatory gaps in digital consent.
- How is ISO/IEC 29100 interoperable with Convention 108?
- How does the OPN Glassbox-model improve these principles?
- It is mentioned that ISO 29184 is out of data but how is this then tackled in the Glassbox model?
- Linking the introduction (e.g., transparency) with the conclusion (e.g., referring to accountability).
- Cultural impoliteness, why is this relevant?
- Consent is sovereign, please clarify this statement at page 8.
- Page 11 starts with a statement that is literally copy pasted. Please contextualise it first and clarify why it is relevant.
- Prioritising transparency can clash with data minimisation too? Please nuance this statement.
- Personal data control removes requirement of data protection? Please nuance this statement. Empowerment and control under the GDPR have a different meaning than pure autonomous data management. Similarly, check 4.2.1: data protection not needed?
- Please explain the statements in section 4.2 as well as 4.3.1 and 4.4. The reader cannot follow this easily at the moment.
- Appendices require additional contextualisation and clarification.

(2) The structure of the paper reads less smoothly. A general description and contextualisation explaining step by step how the paper intends to address the current shortcomings is recommended.

- Clearer introduction of the terms glassbox, governance, meta-privacy model, OPN abbreviation needed.
- The specification of methodology is absent at the moment.
- How does the OPN meta model precisely use the mentioned legislation principles, the mentioned ISO standards, how does it incorporate them?
- What is the difference between normative (2.2.1) and non-normative and how is the line drawn?
- More structured argumentation needed for the important section 2.2.2.1.
- At present, Section 2.2.3 is not helpful in building an argument for the OPN model because it is quite unclear.
- Why is Quebec privacy law mentioned in the end? Introduce it first.

(3) The paper would benefit from describing some shortcomings of the proposed Glass-boxed meta model. This is lacking so far.

- Consider the phenomenon of consent fatigue.
- Point towards further research opportunities.

(4) The source citation could be more precise and many more source citations are needed for various papers to strengthen the arguments.

(5) Please review the layout, as several sections have footnotes that are mixed up or interfere with the readability of the text.

(6) Spelling and grammatical structure needs to be checked again thoroughly. Several typo's are present which delay and complicate the reading of the paper.
- see page 1 abstract, introduction
- see page 5 2.2.3
- see page 6 2.2.3.2 and 2.2.3.3 (re-write sentences, principle = principal?)
- see page 8 2.2.4.
- Please do not write words in capital letters without defining them.

---

### Decision · Program_Chairs · 2025-07-25

Reject

---

> ### Author Response · Authors · 2025-07-25
> **Response**
>
> Hi Reviewers,
>
> Appreciate and agree with the comments, as I put in, almost a first draft, due to time and resources complaints,   recommend requesting an abstract, rather than a finished paper.
>
> The input will help finish this white paper for submission at the next / next year ...